# Peer review of "Revised Annotation and Characterization of Novel Aedes albopictus miRNAs and Their Potential Functions in Dengue Virus Infection"

_biology, 2022, doi:10.3390/biology11101536_

Round 1
Reviewer 1 Report
1.This manuscript is of reference value, but it needs revision.
2.The differences between infected and uninfected miRNA are not clearly discussed.
-
3. Enhanced discussion and new citations are needed
Author Response
Dear Reviewer 1, we compiled all the comments and addressed it in the file attached.
Thank you.

Reviewer 2 Report
Manuscript by Azlan et al applied high-depth small RNA-seq to RNA samples from Aedes albopictus cell line C6/36 and report the finding of novel mature miRNAs, some of them, probably involved in DENV infection. These findings are interesting, however, I have some mayor concerns:
The authors do not make the differentiation between C6/36 and complete mosquito samples and report their findings as follows:
“A total of 110 novel mature miRNAs which derived from 125 precursors were 185 identified. A complete list of both mature and precursor novel miRNAs can be found in 186 Supplementary Info 1. The majority of mature Ae. albopictus miRNAs were found to be 22 187 bp in length, while in the case of precursor miRNAs, most of them ranged between 58-65 188 bp (Figure 2). Ae. albopictus miRNAs varied in terms of genomic loci – 49.9% and 48.2% of 189 them located in the intergenic and intronic regions, while the remaining resided within 190 coding sequence (CDS). “
The expression of novel miRNAs was not confirmed experimentally in mosquito samples, although the possibility of the presence of these miRNAs is very likely, this needs to be corroborated in order to report the findings as “novel Aedes albopictus miRNAs” especially considering the differences between C6/36 cells and native mosquito cells, like the ones cited in the following references (https://doi.org/10.1371/journal.pntd.0000856, doi: 10.1093/gigascience/gix135)
Authors claim “A case study in Singapore reported 270 that patients with DENV1 infection may exhibit more severe illnesses compared to those 271 infected with DENV2 [39]. Besides, different symptoms were observed between DENV1 272 and DENV2-infected patients [39], suggesting that different serotype evokes different 273 molecular responses in humans. Therefore, we infer that different serotype of DENV will 274 elicit different transcriptional responses in Ae. aegypti mosquitoes.”
Is very problematic to generalize an entire serotype in their transmissibility and virulence phenotypes, especially since there is evidence of high variability intraserotype even in the capacity to infect mosquito cells (doi: 10.7554/eLife.42496, https://doi.org/10.1371/journal.ppat.1007187)
Authors claim "In this study, we discovered that our newly annotated Ae. albopictus miRNAs in C6/36 cells were differentially expressed after 3 days post-infection with DENV1. A total of 102 miRNAs were found to be differentially expressed (P-value <0.05), whereby 59 of 278 them were novel miRNAs”
The same limitation, in order to claim the involvement of new miRNAs in Aides albopictus mosquito, needs to be confined experimentally. An alternative is to limit the reach of the claims based on the results and report the findings as new miRNAs in C6/36.
Author Response
Dear Reviewer 2,
Please find the attached file for the response.
Thank you.

Reviewer 3 Report
Brief summary
In this article, Azlan et al. perform an extensive study of micro-RNAs in Aedes albopictus, combining publicly available datasets with their own data generated in C6/36 cells. This study investigates previously known and unknown miRNAs, their production throughout the developmental stages of the mosquito, and their presence following DENV infection. Finally, the possible targets of differentially expressed miRNAs during infection is predicted to inform on their possible role.
General comments
Overall, the article structure and goals are clearly presented. The subject under study is of interest for the scientific community and the results will provide useful information. The methods implemented are clear and appropriate to investigate the subject.
The cited references should be expanded to include the most recent published articles on similar matters: most of the citations are at least 5 years old and much has been done since. First and foremost, a new genome annotation of Ae. Albopictus using PacBio technology was released in 2019 (AalbF2), which included miRNA annotation. Several studies have also been published on small RNA production throughout development and following infection in Aedes mosquitoes.
The discussion needs some re-writing to emphasize the relevance of the results, the gaps that still need investigation and the prospects opened by this study. As for the introduction, more recent literature should be included.
Specific comments
Introduction: as previously stated, more recent information should be included.
Line 121: other miRNAs datasets for Ae. albopictus are available (some more recent). State why this dataset in particular (Batz et al. 2017) has been chosen as reference.
Line 142: “binding required in miRNA positions 2-7”. Confront with line 140 and 201. Why using different seeds?
Line 185: Add a sentence to say how many miRNAs were found in total and how it relates to other studies.
Line 197: “These novel miRNAs shared the same seed region, which, according to miRDeep2, is the first 18 nt of a read sequence” does not sound very accurate. The seed region can be modified, it is up to the researcher to decide if this length is meaningful or not. Also, throughout the manuscript “seed region/sequence” is sometimes 2-7, 2-8, or, in this case, 1-18. Some more clarity is needed.
Figure 3: in the caption we have “Cx. quinquefasciatus”. I suspect it is an oversight.
Figure 4: Higher picture quality might be necessary. It is also unclear to me what each dot represents in the plot. Is it all the miRNAs detected? Or only the novel ones?
Figure 5: possibly better as a supplemental figure.
Line 236: I find the ending blunt. A small paragraph to summarize the findings, the unknowns and the perspectives would provide useful. More of the recent literature should be discussed.
Author Response
Dear Reviewer 3,
Please find the attached file for the response.
Thank you.

Round 2
Reviewer 1 Report
/
Reviewer 2 Report
Updated manuscript by Azlan et al. shows clear improvement.